# Metal-free electrochemical dihydroxylation of unactivated alkenes

Min Liu[1], Tian Feng[1], Yanwei Wang[1], Guangsheng Kou[1], Qiuyan Wang[1] ✉, Qian Wang[1] & Youai Qiu [1] ✉

Herein, a metal-free electrochemical dihydroxylation of unactivated alkenes is described. The transformation proceeds smoothly under mild conditions with a broad range of unactivated alkenes, providing valuable and versatile dihydroxylated products in moderate to good yields without the addition of costly transition metals and stoichiometric amounts of chemical oxidants. Moreover, this method can be applied to a range of natural products and pharmaceutical derivatives, further demonstrating its synthetic utility. Mechanistic studies have revealed that iodohydrin and epoxide intermediate are formed during the reaction process.

Vicinal diols widely exist in natural products, synthetic molecules, and biologically active compounds. They are also key intermediates in synthetic chemistry, so the synthesis of dihydroxyl compounds is of great importance[1–5]. Direct dihydroxylation of readily available alkenes is regarded as a straightforward and promising strategy to access 1,2-diols. Typical oxidative dihydroxylations of alkenes described in chemistry textbooks usually use stoichiometric amounts of chemical oxidants, such as $KMnO_4$ and $OsO_4$ (Fig. 1A). Although these representative methods are powerful, toxic metal salt wastes are always produced in the reactions[6–11]. In recent years, significant developments have been made in this field through transition-metal catalyzed methodologies (e.g., Mn, Ru, Fe, Pd, et al.)[12–21]. However, the direct dihydroxylation of unactivated alkenes is still limited by (i) the employment of toxic noble metals, (ii) the harsh reaction conditions and over-oxidation, (iii) the required stoichiometric amounts of chemical oxidants, and iv) the multistep process. Therefore, there is a great need to develop alternative strategies for dihydroxylation of alkenes.

It is worth highlighting that several methods utilizing photocatalysis[22–25], electrocatalysis[26–30], and electrophotochemistry[31] have been developed to construct 1,2-diols in the past few years. However, the substrates are generally limited to activated alkenes, for example, styrene, and many approaches require the participation of metals. Hence, the metal-free dihydroxylation of unactivated alkenes remains a formidable challenge. With the blossom of electrochemistry[32–82], the direct electrooxidation of alkenes to generate radical cations followed by the attack of $H_2O$ is one of the most

straightforward and effective approaches for synthesizing 1,2-diols. Nevertheless, unactivated alkenes with inherently high oxidation potential often leads to over-oxidation of the reaction, thereby limiting the functional group tolerance of substrates. Alternatively, a mediated electrolysis would be an ideal choice to realize such a transformation. To address this issue, we envisaged the opportunity to develop metal-free dihydroxylation reactions from unactivated alkenes and $H_2O$ under iodine-mediated electrochemical conditions. This was inspired by previously elegant works[83, 84], where active iodine species were generated in situ using electrooxidation, and this strategy successfully bypassed the barrier of high oxidation potentials of unactivated alkenes and avoided the use of transition metal catalysts. To our knowledge, metal-free electrochemical dihydroxylation of unactivated alkenes has yet to be realized.

With our continuous interests in sustainable electrochemistry[85–88], here we report our effort in the development of this metal-free electrochemical dihydroxylation of unactivated alkenes, as a general, direct, and more environmentally friendly approach for the synthesis of aliphatic vicinal diols (Fig. 1B). Salient features of our method include: (a) it is a transition-metal free process, (b) $H_2O$ is used as a green and easily available hydroxyl source, (c) electricity is the oxidative source and replaces the use of stoichiometric amounts of oxidants, and (d) direct dihydroxylation of unactivated alkenes and naturally occurring complex compounds is achieved. Detailed mechanistic insights of this process provided strong support for the presence and key role of an epoxide intermediate.

[1]State Key Laboratory and Institute of Elemento-Organic Chemistry, Frontiers Science Center for New Organic Matter, College of Chemistry, Nankai University, 94 Weijin Road, Tianjin 300071, China. ✉e-mail: qywang@nankai.edu.cn; qiuyouai@nankai.edu.cn

## Results

### Optimization of electrochemical dihydroxylation of unactivated alkenes

We began our investigation into the proposed transformation using 4-(pent-4-en-1-yloxy)−1,1'-biphenyl (**1a**) as a model substrate (Table 1). To our delight, dihydroxylation of unactivated alkene **1a** was achieved in 80% isolated yield under a constant current (50 mA), using water as the hydroxyl source in a mixture solvent of $^t$BuOH and $H_2O$, with carbon felt (CF) as the anode and platinum plate as the cathode (Table 1, entry 1). Next, a variety of electrolytes were examined (entries 2–5). $Et_4NI$ was found to be the most efficient electrolyte for this transformation, whereas the reaction did not proceed when $^nBu_4NBF_4$ was employed as the electrolyte (entry 5). It was demonstrated that both $NH_4I$ and TFA could promote this conversion (entries 6 and 7), and $Et_4NI$ played an essential role as no product could be detected in the absence of $Et_4NI$ (entry 8). We speculated that the addition of $NH_4I$ and TFA could adjust

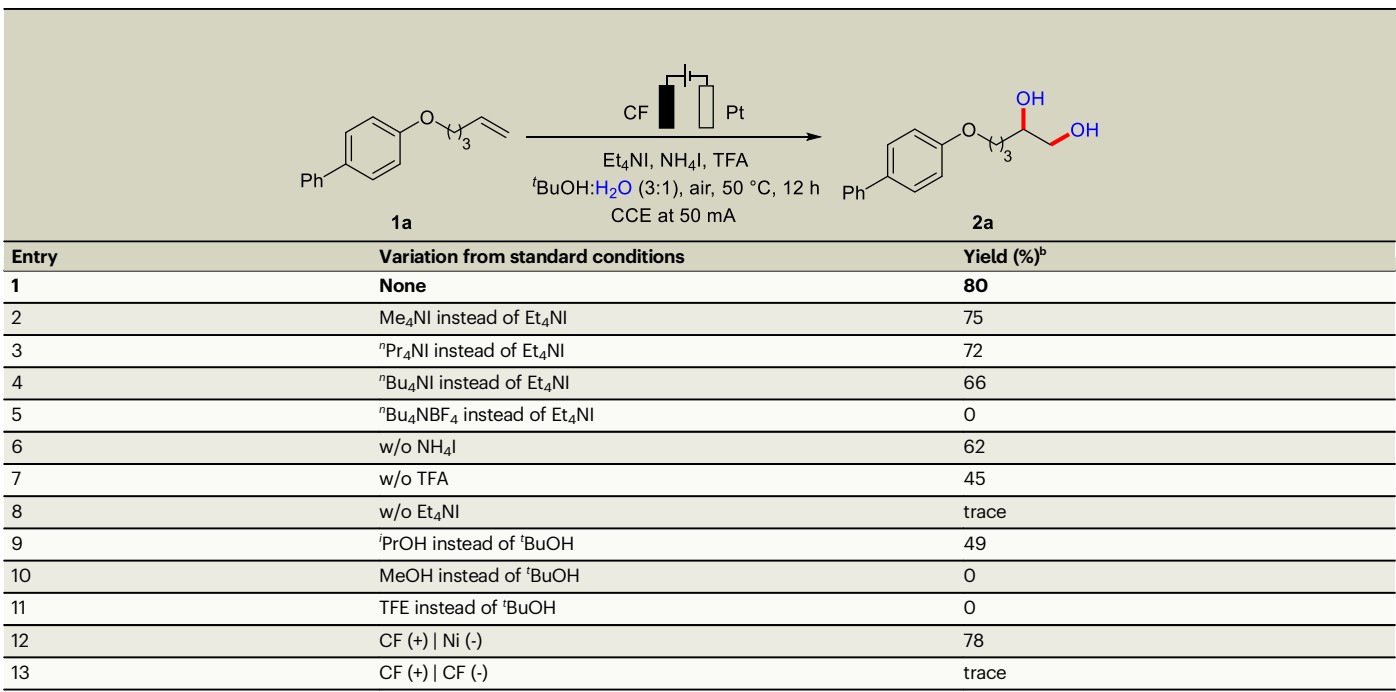

**Fig. 1 | Strategies for the dihydroxylation of alkenes. A** Typical methods for vicinal diols synthesis in chemistry textbooks. **B** This work: metal-free electrochemical dihydroxylation of unactivated alkenes (this work).

**Table 1 | Screening of the reaction conditions.**[a]

| Entry | Variation from standard conditions | Yield (%)[b] |
|---|---|---|
| **1** | **None** | **80** |
| 2 | $Me_4NI$ instead of $Et_4NI$ | 75 |
| 3 | $^nPr_4NI$ instead of $Et_4NI$ | 72 |
| 4 | $^nBu_4NI$ instead of $Et_4NI$ | 66 |
| 5 | $^nBu_4NBF_4$ instead of $Et_4NI$ | 0 |
| 6 | w/o $NH_4I$ | 62 |
| 7 | w/o TFA | 45 |
| 8 | w/o $Et_4NI$ | trace |
| 9 | $^iPrOH$ instead of $^tBuOH$ | 49 |
| 10 | MeOH instead of $^tBuOH$ | 0 |
| 11 | TFE instead of $^tBuOH$ | 0 |
| 12 | CF (+) | Ni (-) | 78 |
| 13 | CF (+) | CF (-) | trace |

Bold formatting shows that entry 1 is the optimal reaction condition.
[a]Reaction conditions. **1a** (0.3 mmol), $Et_4NI$ (2.0 equiv.), $NH_4I$ (2.0 equiv.), TFA (3.0 equiv.), $^tBuOH$ (3.0 mL) and $H_2O$ (1.0 mL) under 50 mA constant in an undivided cell at 50 °C for 12 h with carbon felt (CF) as anode and Pt plate as cathode.
[b]Isolated yield. TFA = trifluoroacetic acid; TFE = 2,2,2-trifluoroethanol.

the PH value of the reaction system, which is of great significance to the transformation, especially in the proposed mechanistic steps from an epoxide intermediate to the diol product. The effect of the solvent was also investigated, and a combination of $H_2O$ with $^iPrOH$, MeOH, or TFE was tested (entries 9–11). $^tBuOH$ and $H_2O$ as a mixed solvent was found to be optimal for this transformation. Using a Ni plate instead of a Pt plate as the cathode provided a slightly decreased yield of the desired product (78%) (entry 12). When a CF was used as the cathode instead of a Pt plate, the yield of **2a** dropped significantly to only a trace amount (entry 13).

## Substrate scope

With the optimal conditions in hand, we then explored the scope of alkenes (Fig. 2). The effect of the length of the carbon chain of the alkene on the reaction was examined first. To our delight, the substrates with a varying carbon chain length from 1 to 8 provided the corresponding dihydroxylated products in moderate to good yields (**2a**–**2h**, 52–80%). Next, we also examined the electronic effects on the reaction. The *p*-tolyl-containing substrate could undergo the reaction smoothly and provide the corresponding product **2i** (78%). Comparable results were obtained when the *para*-position of the phenyl ring

**Fig. 2 | Scope of the dihydroxylation of unactivated alkenes.** [a]Reaction condition A: **1** (0.3 mmol), Et₄NI (2.0 equiv.), NH₄I (2.0 equiv.), TFA (3.0 equiv.), $^tBuOH$ (3.0 mL), and $H_2O$ (1.0 mL) under 50 mA constant in an undivided cell at 50 °C for 12 h with carbon felt (CF) as anode and Pt plate as cathode.[b] For details, please see the Supplementary Information.

was substituted with an F or Cl (**2j, 2k**). Interestingly, substrates with other electron-withdrawing groups in the *para*-position, such as (Br, CF₃, and CN), resulted in the corresponding dihydroxylated products in lower yields (**2l–2n**). Alternatively, when different substituent groups were attached to the *meta*- and *para*-positions of the benzene ring, the corresponding product **2o** could be obtained in moderate yield. It is important to note that this transformation is not restricted to electrochemical sensitive groups. For example, substrates bearing a ketone or ester group furnished the corresponding products in good yields (**2p, 2q**). To our delight, the reaction also proceeded smoothly when the branched chain of the alkenes was attached to the benzene ring without oxygen atoms, giving the corresponding dihydroxylated product **2r** in an isolated yield of 66%. Besides, alkenes containing alkyl ester motif could be tolerated in this transformation, and gave the corresponding product in 63% (**2s**). Then, we also realized the dihydroxylation reaction of heteroarenes such as thiophene and pyrazole derivative under standard conditions to give the corresponding product **2t** (54%) and **2u** (63%). Encouragingly, some polysubstituted alkenes were also able to react smoothly, and the corresponding dihydroxylated products (**2v–2z**) were obtained in

moderate yields and with diastereomeric ratio of 1:1 for **2x**, respectively. Notably, this protocol was also found to effectively functionalize a range of cyclic and linear aliphatic alkenes. When a cyclohexene motif was introduced, the corresponding cyclohexanediol product could be obtained in good yield (**2 y** and **2z**, 83% and 86%) with excellent regioselectivity (d.r. > 20:1). Moreover, **2 y** was verified by the analysis of single-crystal X-ray diffraction (CCDC-2286663). As for linear alkenes, different number of the carbon chains could furnish the corresponding products in moderate to good yields, the efficiency increased with the growth of the carbon chain (**2aa–2ad**). Moreover, this transformation was tolerant of aliphatic alkenes with halides such as Cl, Br, and some electrochemical sensitive groups such as NHTs and ester, all the corresponding products were obtained in good yields (**2ae–2ai**).

Notably, due to the importance of diols in natural products, pharmaceuticals, and bioactive compounds, this protocol was further extended to the dihydroxylation of complex compounds. Unactivated alkenes derived from Oxaprozin (**1aj**), Menthol (**1ak**), Ibuprofen (**1al**), Flurbiprofen (**1am**), and Lithocholic (**1an**) Formononetin (**1ao**), Gemfibrozil (**1ap**), Sulbactam acid (**1aq**), Artemisinin (**1ar**) were all suitable for this electro-oxidative dihydroxylation system and provided the desired dihydroxylated products in moderated yields. These results showed that this strategy could offer a potential synthetic application to the introduction of vicinal hydroxyl functional groups for complex natural compounds and drug derivatives.

To our delight, we found that by adjusting the experimental conditions (for details, please see the Fig. 3 of Supplementary

Information), a series of epoxide products could be obtained, as shown in Fig. 3. The alkene scope was explored using the optimal reaction conditions for the generation of epoxides. Firstly, we examined the effect of the electronic properties of the substrate substituents on the reaction yield. When an electron-donating group such as methyl was in the *para*-position on the phenyl ring, the corresponding product could be obtained in 78% (**3b**). In contrast, substrates with electron-withdrawing groups (e.g., F, Br, CF₃) gave the corresponding epoxidation products in significantly lower yields (**3c–3e**, 42–63%). In addition, an alkene derived from naphthalene reacted smoothly and gave the product **3f** in 31% yield with a considerable amount of the starting material remaining. When different substituent groups were in the *meta*- and *para*-positions on the phenyl ring, we obtained the corresponding product **3g** in moderate yield. Substrates with different alkyl chain lengths afforded the corresponding product in moderate to good yields (**3h–3k**, 66–86%). Furthermore, an *ortho*-phenyl substituted substrate also furnished the desired product in 52% isolated yield (**3l**). Encouragingly, some multi-substituted alkenes were also able to react successfully, and the epoxide products were obtained in moderate yields (**3 m, 3n**). Moreover, the transformation proceeded smoothly with a linear alkene containing a chloride and an ester group, which generated the desired product in 45% yield (**3o**).

## Mechanistic studies

To further illustrate the synthetic usage of this method, we carried out a gram-scale reaction (Fig. 4A). The reaction of **1a** (5 mmol) under slightly modified conditions gave diol **2a** in a 78% isolated yield (1.06 g) after 18 h. To gain insights into the mechanism of this transformation, we next carried out several control experiments. Firstly, isotope labeling experiments showed that the hydroxy group of product **2a** was from water (Fig. 4B, a, detected by high-resolution mass spectrometry (HRMS), for details, please see the Fig. 5 of Supplementary Information). Next, we speculated that iodine anion could generate molecular iodine under anodic oxidation, as a consequence, iodine was used to replace Et₄NI and NH₄I. The mixture of **4** and **4'** was obtained in 30% yield without electricity (Fig. 4B, b), indicating that the formation of iodine during the reaction process could react with alkenes and the corresponding iodide alcohols might be the intermediates. In addition, under the standard conditions, **4** and **3i** were also obtained in 5% and 12%, respectively, whereas the other iodide alcohol **4'** was not detected, demonstrated that **4** and **3i** might be the key intermediates of this transformation (Fig. 4B, c). Next, a series of control experiments using **4** as starting material were conducted. Under condition A, **2a** could be obtained in 82% from **4**, while no **2a** was detected without electricity (Fig. 4B, d, e), denoted that electric current played an essential role in the formation of **2a**. Significantly, it was detected that **2a** could not be generated without the addition of

**Fig. 3 | Scope of the epoxidation of unactivated alkenes.** ᵃReaction condition B: **1** (0.3 mmol), Et₄NI (2.0 equiv.), NH₄I (2.0 equiv.), TFA (3.0 equiv.), DMI (3.0 mL) and H₂O (1.0 mL) under 50 mA constant in an undivided cell at 50 ℃ for 12 h with

carbon felt (CF) as anode and Pt plate as cathode. DMI = 1,3-dimethyl-2-imidazolidinone.

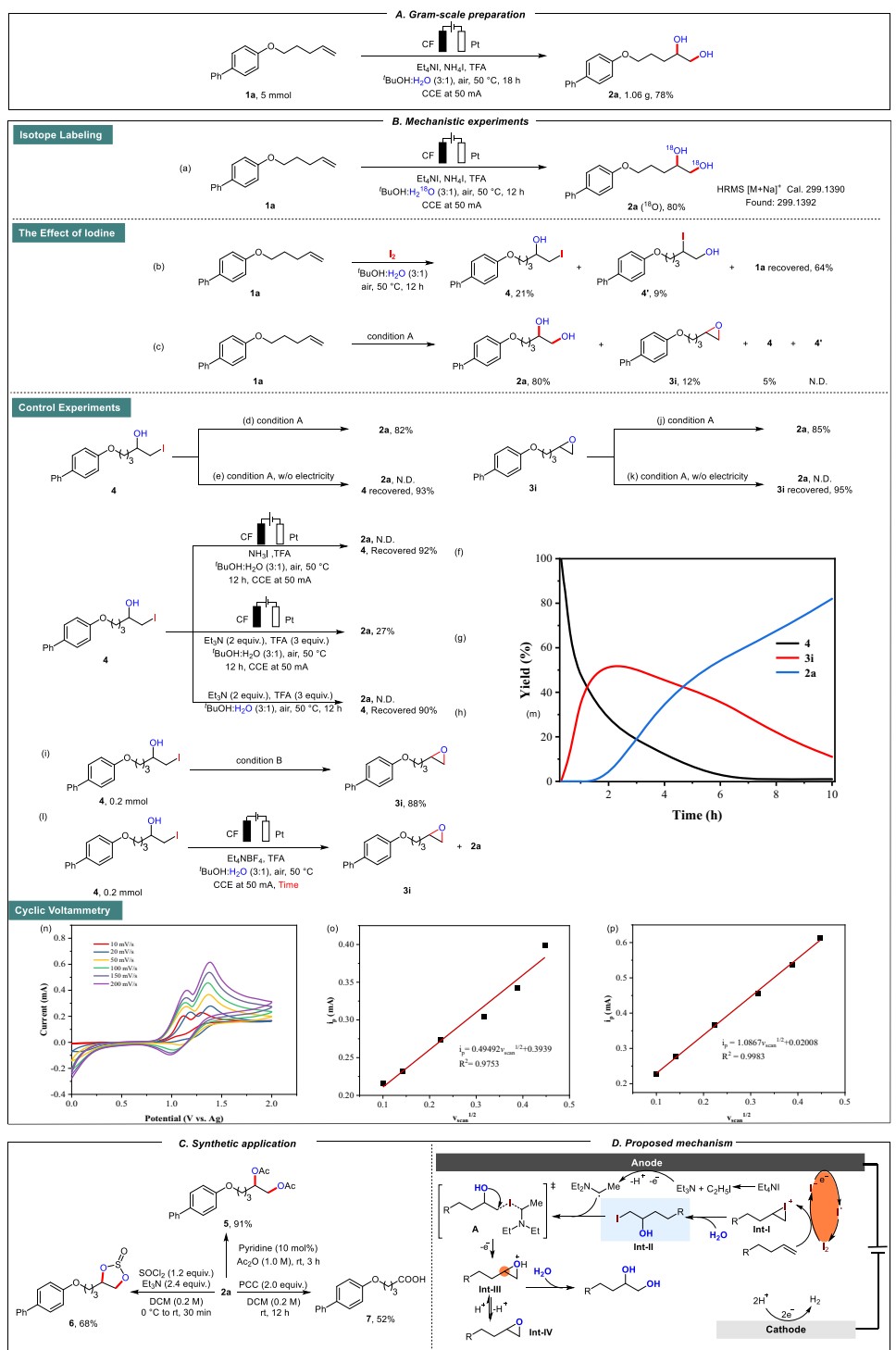

**Fig. 4 | Gram-scale preparation and control experiments. A** The compound **2a** could be prepared in gram-scale with higher isolated yield. **B** Mechanistic experiments Cyclic voltammetry, using glass carbon as working electrode, Pt plate was used as counter electrode, and an Ag wire was used as the reference electrode in acetonitrile. (CH$_3$CN, Et$_4$NI (10.0 mM), 0.1 M Et$_4$NBF$_4$, 50 mVs$^{-1}$). **C** Synthetic application. **D** Proposed mechanism.

Et$_4$NI during the screening of the condition reaction (Fig. 4B, f). And we have conducted GC-MS detection of the reaction solution and detected the presence of Et$_3$N in the system, which then underwent electrooxidation to generate α-aminoalkyl radicals. Notably, α-aminoalkyl radicals have the ability to extract iodine and bromine atoms from organic compounds. Thus, we speculated that electricity plays a role in the oxidation of triethylamine generated in-situ in our reaction system giving α-aminoalkyl radicals for the activation of carbon-iodine bonds. Consequently, some control experiments were performed in the presence of Et$_3$N with/without electricity (Fig. 4B, g, h), and the results turned out that **2a** could be obtained under electrochemical conditions assisted by tertiary amine. However, no product was detected without electricity. These results confirmed that tertiary amines could promote this transformation under electrochemical conditions. Then, **4** could generate epoxy product **3i** under condition B in 88% (Fig. 4B, i), which could furnish **2a** in 85% under condition A, whereas no **2a** was detected without electricity with 95% of **3i** recovery (Fig. 4B, j, k). These results also showed electric current was pivotal to this reaction. Then,

in order to further prove the conversion relationship between **4, 3i**, and **2a**, under a current of 50 mA, the curves of the yields of **4, 3i**, and **2a** were plotted as a function of reaction time (Fig. 4B, l). Initially, **3i** showed an increasing trend with the decreasing of **4**. After **2** h, **3i** turned over to decrease with the formation of **2a**. This confirmed that **4** continuously generated epoxy product **3i** which then transformed to **2a** during the reaction process (Fig. 4B, m). Notably, we performed Cyclic voltammetry (CV, for details, please see the Supplementary Information) tests of Et₄NI with variable scan rates (Fig. 4B, n), and the linear fit analysis provided support for the possibility that a diffusion control process might be involved in this conversion (Fig. 4B, o, p).

In addition, several experiments were carried out to showcase the utility of the products (Fig. 4C). Firstly, hydroxy groups could be protected by acetic anhydride in excellent efficiency, gave the corresponding product **5** in 91%. Next, the vicinal hydroxy groups could react with thionyl chloride under basic conditions to generate product **6** in 68%. In addition, **2a** could also be transformed into the corresponding carboxylic acid product **7** in moderated yield by the simple oxidation of pyridinium chlorochromate.

According to the control experiments and literature reports[89–95], a plausible mechanism was proposed (Fig. 4D). Firstly, electrochemical oxidation of I⁻ generated iodine radical, which then dimerized to generate iodine. Then, iodine reacted with alkene to generate iodonium intermediate **Int-I**. **Int-I** tended to engage in the nucleophilic addition with water to generate intermediate **Int-II**. Meanwhile, Et₄NI was decomposed into Et₃N on the anode, which was then electrooxidized to obtain the α-amino radical. Then, the α-amino radical was capable of abstracting iodine atom from alkyl iodides (intermediate **Int-II**) to give **Int-III** via intramolecular cyclization. Simultaneously losing an electron during this process. Subsequently, a ring-opening reaction occurred of **Int-III** by the attack of H₂O furnishing the products. Simultaneously, an equilibrium reaction between **Int-III** and free epoxy intermediate (**Int-IV**) proceeded in this transformation.

## Discussion

In conclusion, a metal-free electrochemical dihydroxylation of unactivated alkenes is reported. The transformation proceeds smoothly under mild conditions with a wide range of aliphatic unactivated alkenes, providing valuable dihydroxylated products with good efficiency. This synthetic method achieves the dihydroxylation of unactivated alkenes without the addition of toxic and expensive transition metals or stoichiometric amounts of oxidants. In addition, dihydroxylation reactions of natural products and pharmaceutical derivatives have further demonstrated the synthetic utility of this transformation. Detailed control experiments and mechanistic studies give strong, supportive evidence of the proposed mechanism. Further applications of electrocatalytic difunctionalization of unactivated alkenes are currently underway in our laboratory.

## Methods

### General procedure for metal-free electrochemical dihydroxylation of unactivated alkenes

The electrocatalysis was carried out in an undivided cell with a carbon felt (CF) anode (5.0 mm × 10.0 mm × 20.0 mm) and a Pt plate cathode (0.1 mm × 10.0 mm × 20.0 mm). To a 15 mL oven-dried undivided electrochemical cell equipped with a magnetic bar were added primary unactivated alkenes (0.3 mmol, 1.0 equiv.), Et₄NI (154.2 mg, 0.6 mmol, 2.0 equiv.), NH₄I (86.9 mg, 0.6 mmol, 2.0 equiv.) and TFA (67.0 μL, 0.9 mmol, 3.0 equiv.). Then ᵗBuOH (3 mL) and H₂O (1 mL) were added under air. The electrocatalysis was performed at 50 °C with a constant current of 50 mA maintained for 12 h. The electrodes were washed with EtOAc (3 × 5 mL) in an ultrasonic bath. Then the solution of Na₂S₂O₃ (10 mL) was added to the system, and the resulting mixture was extracted with EtOAc (2 × 20 mL). The combined organic phase was dried with anhydrous Na₂SO₄, filtered, and concentrated in vacuo. The crude product was purified by column chromatography to furnish the desired product.

### General procedure for electrochemical epoxidation of unactivated alkenes

The electrocatalysis was carried out in an undivided cell with a carbon felt (CF) anode (5.0 mm × 10.0 mm × 20.0 mm) and a Pt plate cathode (0.1 mm × 10.0 mm × 20.0 mm). To a 15 mL oven-dried undivided electrochemical cell equipped with a magnetic bar were added primary unactivated alkenes (0.3 mmol, 1.0 equiv.), Et₄NI (154.2 mg, 0.6 mmol, 2.0 equiv.), and NH₄I (86.9 mg, 0.6 mmol, 2.0 equiv.). Then DMI (3 mL) and H₂O (1 mL) were added under air. The electrocatalysis was performed at 50 °C with a constant current of 50 mA maintained for 12 h. The electrodes were washed with EtOAc (3 × 5 mL) in an ultrasonic bath. H₂O (20 mL) was added to the system, and the resulting mixture was extracted with EtOAc (2 × 50 mL). The combined organic phase was dried with anhydrous Na₂SO₄, filtered, and concentrated in vacuo. The crude product was purified by column chromatography to furnish the desired product.

### Gram-scale synthesis of 2a

The electrocatalysis was carried out in an undivided cell with a carbon felt (CF) anode (5.0 mm × 25.0 mm × 50.0 mm) and a Pt plate cathode (0.25 mm × 25.0 mm × 50.0 mm). To an oven-dried undivided electrochemical cell (diameter: 40 mm; length: 130 mm; volume: 200 mL) equipped with a magnetic bar were added primary unactivated alkenes. (5.0 mmol, 1.0 equiv.), Et₄NI (2.57 g, 10.0 mmol, 2.0 equiv.), NH₄I (1.45 g, 10.0 mmol, 2.0 equiv.) and TFA (1.12 mL, 15.0 mmol, 3.0 equiv.). Then ᵗBuOH (60 mL) and H₂O (20 mL) were added under air. The electrocatalysis was performed at 50 °C with a constant current of 50 mA maintained for 12 h. The electrodes were washed with EtOAc (3 × 20 mL) in an ultrasonic bath. Then the solution of Na₂S₂O₃ (50 mL) was added to the system, and the resulting mixture was extracted with EtOAc (2 × 100 mL). The combined organic phase was dried with anhydrous Na₂SO₄, filtered, and concentrated in vacuo. The product was purified by column chromatography to provide 1.06 g (78%) of compound **2a**.

## Data availability

Materials and methods, optimization studies, experimental procedures, mechanistic studies, ¹H NMR, ¹³C NMR, and ¹⁹F NMR spectra, and high-resolution mass spectrometry data are available in the Supplementary Information. The 2y data generated in this study have been deposited in the Cambridge Crystallographic Data Center (CCDC) database under accession code 2286663 [http://www.ccdc.cam.ac.uk/].

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

## Acknowledgements

Financial support from the National Key R&D Program of China (2022YFA1503200), the Fundamental Research Funds for the Central

Universities (No. 63223007), National Natural Science Foundation of China (Grant No. 22371149, 22188101), Frontiers Science Center for New Organic Matter, Nankai University (Grant No. 63181206) and Nankai University are gratefully acknowledged.

## Author contributions

Y.Q. directed the project. Y.Q., M.L., and Q.-Y.W. conceived and designed the study and wrote the manuscript. M.L., T.F., Y.W., G.K., Q.-Y.W., and Q.W. performed the experiments, mechanistic studies, analyzed the data, and revised the manuscript. All authors contributed to scientific discussion.

## Competing interests

The authors declare no competing interests.
