## [Peer Review File · Nature Communications]

REVIEWER COMMENTS

Reviewer #1 (Remarks to the Author):

Recommendation: Publish after minor revisions.

Comments:

The authors describe a metal-free electrochemical dihydroxylation of unactivated alkenes. The scope of the reaction is evaluated showing good functional group tolerance. Vicinal diols widely exist in natural products, and also are key intermediates in synthetic chemistry. Although several methods for the difunctionalization of alkenes utilizing electrocatalysis (ref. 26-30 in the manuscript) have been reported, this work developed by Qiu and coworkers has made great advancement and achievement for the dihydroxylation of unactivated alkenes, using water as -OH source in the electrolyte system. Further functionalization of products was reported, and a detail mechanism was proposed. However, I have some scientific concerns should be addressed before I would recommend accepting this manuscript for publication. Additionally, there are some apparent errors in the text and supplementary information that should be corrected as well, which are listed below:

1) The manuscript was sufficiently well written to allow it to be understood easily for review, but still need attention to correct some grammatical and wording. Some examples are given below, but this is not an exhaustive list and I recommend the whole text be checked carefully:

-line 15 – change ‘intermediator’ into ‘intermediate’

-line 25-26 – change ‘the toxic loading of noble metals’ into ‘the employment of toxic noble metals’

-line 83 – change ‘product’ into ‘products’

-line 115-116 – change ‘Oxaprozin (2ag), Menthol (2ah), Ibuprofen (2ai), Flurbiprofen (2aj), and Lithocholic (2ak) Formononetin (2al)’ into ‘Oxaprozin (1ag), Menthol (1ah), Ibuprofen (1ai), Flurbiprofen (1aj), and Lithocholic (1ak) Formononetin (1al)’

-line 139 – change ‘2 0’ into ‘2.0’

-line 153 – change ‘form’ into ‘from’

2) The structure of 3l provided in Scheme 3 was not consistent with the description in line 131 of the text. Please check carefully.

3) The HR-MS data of 2a(18O) should be shown in the isotope labeling experiment of the text.

- 4) In the experiment on effect of iodine in the text, 4 and 4' could not be separated respectively, the 4 in control experiments was a sole isomer or a mixture with 4'?
- 5) Whether some oxidative byproducts of 2 or 4 , such as ketone or aldehyde, were detected under the standard conditions?
- 6) What is the meaning of "electronically sensitive groups" in the text? nucleophilic or electrophilic?
- 9) In page S-4 of supplementary information, change 'addition' into 'additive'
- 10) The 'h' into 'hours' should be consistent in the Supplementary Information.
- 11) The 100 MHz and 101 MHz of ¹³C NMR were both used thorough the supplementary information. These should be kept in consistency.
- 12) There were some errors of NMR data in supplementary information, especially number of significant figures in ¹³C NMR and ¹⁹F data. Please check carefully.
- 13) The HR-MS data of 2j, 2k, 2n, 2x and 2ah were invalid and the HR-MS data of 5 should be provided in supplementary information.

Reviewer #2 (Remarks to the Author):

This manuscript reports a radical initiated C(sp³)-H bond oxidative functionalization of alkyl nitriles through a KI mediated indirect anodic oxidation. High chemoselectivities and good stereoselectivities of the reactions could be achieved under the metal-free, external chemical oxidant-free conditions. A reliable mechanism is proposed after some control experiments and cyclic voltammetry experiments were conducted. The results described by the authors are informative and interesting to Nature Communications readers, particularly those working on the electroorganic chemistry. Based on the content and quality of this paper, I think this work is suitable for Nature Communications however some issues should be addressed before the accepting this manuscript for publication.

1. In Scheme 4(c), compound 4' cannot be observed under electrochemical conditions (condition a). The author should give some corresponding explanation for this regioselectivity?
2. The result of Scheme 4(e) shows that the conversion of compound 4 to product 2 requires electricity. In the mechanism description, the author did not explain why electricity is needed.

3. Why does this reaction require two types of iodized salts? Can the amount of iodide be reduced?

4. How to react under argon gas conditions?

Reviewer #3 (Remarks to the Author):

The authors present electrochemical methods for the dihydroxylation and epoxidation of alkylalkenes, which employ mild yet effective reaction conditions for transforming simple materials into value-added products. Intriguingly, the reactions selectively yield diols or epoxides through a mere alteration of the conditions. The publication of this work is recommended, provided the following concerns are addressed:

A. For products originating from internal alkenes, kindly specify the stereochemistry of the predominant product. Additionally, clarify if cis- and trans-alkenes yield the same or different products.

B. The experiments demonstrate that electricity is required to convert intermediate 4 into the desired products. However, the proposed mechanism does not account for this observation. Please revise the mechanism accordingly to incorporate this crucial aspect.

Comments from the reviewers:

Reviewer 1

Question 1: line 15 – change ‘intermediator’ into ‘intermediate’

Response: Thanks for the comments. We have revised ‘intermediator’ in line 15 into ‘intermediate’ in the manuscript.

line 25-26 – change ‘the toxic loading of noble metals’ into ‘the employment of toxic noble metals’

Response: Thanks for the comments. ‘the toxic loading of noble metals’ in line 25-26 has been replaced by ‘the employment of toxic noble metals’ in the manuscript.

-line 83 – change ‘product’ into ‘products’

Response: Thanks for the comments. We have revised ‘product’ in line 83 into ‘products’ in the manuscript.

-line 115-116 – change ‘Oxaprozin (**2ag**), Menthol (**2ah**), Ibuprofen (**2ai**), Flurbiprofen (**2aj**), and Lithocholic (**2ak**) Formononetin (**2al**)’ into ‘Oxaprozin (**1aj**), Menthol (**1ah**), Ibuprofen (**1ai**), Flurbiprofen (**1aj**), and Lithocholic (**1ak**) Formononetin (**1al**)’

Response: Thanks for the comments. We have revised Oxaprozin (**2ag**), Menthol (**2ah**), Ibuprofen (**2ai**), Flurbiprofen (**2aj**), and Lithocholic (**2ak**) Formononetin (**2al**)’ in line 115-116 into ‘Oxaprozin (**1aj**), Menthol (**1ah**), Ibuprofen (**1ak**), Flurbiprofen (**1al**), and Lithocholic (**1am**) Formononetin (**1an**)’ in the manuscript. To demonstrate the good generality of the reaction, we added several substrates, so we changed the substrate number and highlighted it in yellow, and the corresponding data have been added to the manuscript and Supplementary information.

-line 139 – change ‘2 0’ into ‘2.0’

Response: Thanks for the comments. We have revised ‘2 0’ into ‘2.0’ in line 139 in the manuscript.

-line 153 – change ‘form’ into ‘from’

Response: Thanks for the comments. The spelling of ‘form’ has been corrected to ‘from’ in line 153.

Question 2 The structure of **31** provided in Scheme 3 was not consistent with the description in line 131 of the text. Please check carefully.

Response: Thanks for the comments. We have revised the description of **31** in line 131 into ‘ortho-phenyl’ in the manuscript.

Question 3 The HR-MS data of **2a**(¹⁸O) should be shown in the isotope labeling experiment of the text.

Response: Thanks for the comments. We have added the HR-MS data ([M+Na]⁺ Found: 299.1392) of **2a** (¹⁸O) to the manuscript (Fig. 4(a)).

Question 4 In the experiment on effect of iodine in the text, **4** and **4'** could not be separated respectively, the **4** in control experiments was a sole isomer or a mixture with **4'**?

Response: Thanks for the comments. We have monitored the reaction by TLC, and intermediates **4** and **4'** were detected simultaneously in the first 4.5 h under conditions A, but the proportion of **4'** during the reaction was less than **4**, and **4'** was barely detected after 4.5 h. In addition, we have plotted curves of the yields of **4** and **4'** with time (**Fig.R1**). Furthermore, we chose 1 h and 2.5 h to purify the reaction system and determined the ratio of **4** to **4'** (78:24 (1 h), 66:30 (2.5 h)) by ¹H NMR.

And the intermediate **4** employed in control experiments was a sole isomer. Conclusively, **4** was a main intermediate in the reaction system, so we selected **4** to conducted a series of control experiments.

Fig. R1

¹H NMR of Compounds 4 and 4' (1 h) (400 MHz, CDCl₃):

¹H NMR of Compounds 4 and 4' (2.5 h) (400 MHz, CDCl₃):

Question 5 Whether some oxidative byproducts of **2** or **4**, such as ketone or aldehyde, were detected under the standard conditions?

Response: Thanks for the comments. We chose several substrates (**1a**, **1b**, **1c**) to monitor the reaction system, and only substrates, intermediates (iodides **4**, epoxides) and products were detected. No oxidative byproducts were detected under the standard conditions.

Question 6 What is the meaning of “electronically sensitive groups” in the text? nucleophilic or electrophilic?

Response: Thanks for the comments. We are so sorry for the mistake, and the phrase of “electronically sensitive groups” should be revised in “electrochemical sensitive groups”. We have done the revision in the manuscript. The phrase “electrochemical sensitive groups” means the functional groups which are easily oxidized or reduced under electrochemical conditions, such as aldehyde, ketone, or ester groups. In this reaction system, the substrates with electrochemical sensitive groups, such as carbonyl

(2p), ester (2q) or carbon-halogen(2j-2l) bonds could be tolerant under the standard conditions.

Question 9 In page S-4 of supplementary information, change ‘addition’ into ‘additive’

Response: Thanks for the comments. We have revised ‘addition’ into ‘additive’ in page S-4 of supplementary information.

Question 10 The ‘h’ into ‘hours’ should be consistent in the Supplementary Information.

Response: Thanks for the comments. We have changed ‘hours’ to ‘h’ in the whole supplementary information.

Question 11 The 100 MHz and 101 MHz of ^{13}C NMR were both used thorough the supplementary information. These should be kept in consistency.

Response: Thanks for the comments. We have changed ‘101 MHz’ to ‘100 MHz’ in the whole supplementary information.

Question 12 There were some errors of NMR data in supplementary information, especially number of significant figures in ^{13}C NMR and ^{19}F data. Please check carefully.

Response: Thank you for the comments. We have carefully checked through all of NMR data again, and corrected number of significant figures in ^{13}C NMR and ^{19}F data. The relevant results have been added to the supplementary information, which were marked in yellow.

Question 13 The HR-MS data of **2j**, **2k**, **2n**, **2x** and **2ah** were invalid and the HR-MS data of **5** should be provided in supplementary information.

Response: Thank you for the comments. We have retested the HR-MS data of compounds **2j** ($[\text{M}+\text{Na}]^+$: 237.0897, found: 237.0899), **2k** ($[\text{M}+\text{Na}]^+$: 253.0602, found: 253.0603), **2n** ($[\text{M}+\text{Na}]^+$: 244.0944, found: 244.0947), **2z** ($[\text{M}+\text{Na}]^+$: 321.1461, found: 321.1466), and **2ak** ($[\text{M}+\text{Na}]^+$: 295.1880, found: 295.1881) and tested the HR-MS for

compound **5** ($[M+H]^+$: 357.1697, found: 357.1707). The original spectra were provided as below. To demonstrate the good generality of the reaction, we added several substrates, so we changed the substrate number and highlighted it in yellow in our manuscript.

The HR-MS data for compounds **2j**

The HR-MS data for compounds **2k**

The HR-MS data for compounds 2n

The HR-MS data for compounds 2z

The HR-MS data for compounds 2ak

The HR-MS data for compounds 5

5.70e4

Reviewer 2

Question 1: In Scheme 4(c), compound **4'** cannot be observed under electrochemical conditions (condition a). The author should give some corresponding explanation for this regioselectivity?

Response: Thanks for the comments. We have monitored the reaction by TLC, and intermediates **4** and **4'** were detected simultaneously in the first 4.5 h under conditions A, but the proportion of **4'** during the reaction was less than **4**, and **4'** was barely detected after 4.5 h. So **4'** was completely converted, and no **4'** was detected in the reaction system after 12 h. In addition, we have plotted curves of the yields of **4** and **4'** with time (**Fig. R1**) to illustrate the status of intermediates **4** and **4'** in the reaction system. Furthermore, we chose 1 h and 2.5 h to purify the reaction system and determined the ratio of **4** to **4'** (78:24 (1 h), 66:30 (2.5 h)) by NMR. Conclusively, **4** was a main intermediate in the reaction system, so we selected **4** to conducted a series of control experiments.

Fig. R1

$^1\text{H NMR}$ of Compounds 4 and 4' (1 h) (400 MHz, CDCl_3):

$^1\text{H NMR}$ of Compounds 4 and 4' (2.5 h) (400 MHz, CDCl_3):

Question 2. The result of Scheme 4(e) shows that the conversion of compound **4** to product **2** requires electricity. In the mechanism description, the author did not explain why electricity is needed.

Response: Thanks for the comments. We have conducted GC-MS analysis of the reaction solution and detected the presence of Et₃N in the system (**Fig.R2**). And relevant literature (*J. Am. Chem. Soc.*, **145**, 10967–10973(2023)) has proved that electrolytes such as Et₄NI and n-Bu₄NI could be decomposed into tertiary amine under electrochemical conditions, which then underwent electrooxidation to generate α -amino alkyl radicals. Notably, α -amino alkyl radicals have the ability to extract iodine and bromine atoms from organic compounds (*Science* **367**, 1021–1026 (2020)). Thus, we speculated that electricity plays a role in the oxidation of triethylamine generated in-situ in our reaction system giving α -amino alkyl radicals for the activation of carbon-iodine bonds.

Fig. R2 GC-MS analysis of samples from standard conditions.

Moreover, several control experiments were conducted to confirm our conjecture (**Fig.R3**). Control experiments were performed in the presence of Et₃N or DIPEA

with/without electricity, and the results turned out that **2a** could be obtained under electrochemical conditions with assisted by tertiary amine. However, no product was detected without electricity. These results confirmed that tertiary amines could promote this transformation under electrochemical conditions. In addition, we have added this control experiment to page S17 to the supplementary information.

Fig.R3

According to our experiments results and literatures, we refine the reaction mechanism (**Fig. R4**). Firstly, electrochemical oxidation of I⁻ generated iodine radical, which then dimerized to generate iodine. Then, iodine reacted with alkene to generate iodonium intermediate **Int-I**. **Int-I** tended to engage in the nucleophilic addition with water to generate intermediate **Int-II**. Meanwhile, Et₄NI was decomposed into Et₃N on the anode, which was then electrooxidized to obtain the α -amino radical. Then, the α -amino radical was capable of abstracting iodine atom from alkyl iodides (intermediate **Int-II**) to give **Int-III** via intramolecular cyclization. Subsequently, a ring-opening reaction occurred of **Int-III** by the attack of H₂O furnishing the products. Simultaneously, an equilibrium reaction between **Int-III** and free epoxy intermediate (**Int-IV**) proceeded in this transformation.

Fig. R4 Proposed Mechanism

Question 3. Why does this reaction require two types of iodized salts? Can the amount of iodide be reduced?

Response: Thanks for the comments. Firstly, Control experiments showed that the target product cannot be obtained without Et_4NI in the system. Moreover, we have conducted GC-MS detection of the reaction solution and detected the presence of Et_3N in the system (**Fig. R2**). And relevant literature (*J. Am. Chem. Soc.*, **145**, 10967–10973(2023)) has proved that electrolytes such as Et_4NI and $n\text{-Bu}_4\text{NI}$ could be decomposed into tertiary amine under electrochemical conditions, which then underwent electrooxidation to generate α -amino alkyl radicals. Notably, α -amino alkyl radicals have the ability to extract iodine and bromine atoms from organic compounds (*Science* **367**, 1021–1026 (2020)). Thus, we speculated that electricity participated in the oxidation of triethylamine which generated in-situ in our reaction system to give α -amino alkyl radicals for the activation of carbon-iodine bonds.

Fig. R2 Control experiments and GC-MS analysis of samples from standard conditions.

Furthermore, we screened the effect of the amount of iodized salt on the reaction (**Table R1**). We found that as the amount of iodized salt gradually increased, the yield of the target product also increased (entry 1-5). In addition, when only Et₄NI (2 equiv) existed in the system, product **2a** could be generated in 62% (entry 5). The above results proved that Et₄NI played an essential role as no product could be detected in the absence of Et₄NI (entry 5). However, when only NH₄I (2 equiv) was present in the system, no product could be detected. Furthermore, when we replaced NH₄I with other iodized salts, such as LiI, we could also get the target product in 75% yield (entry 7). The above results indicated that NH₄I was mainly used to offer iodide ions and played a role in improving the yield of reaction.

Table R1

Entry	Variation	Yield ^b (%) of 2a
1	Et ₄ NI (0.5 eq.), NH ₄ I (0.5 eq.)	27
2	Et ₄ NI (1.0 eq.), NH ₄ I (1.0 eq.)	45
3	Et ₄ NI (1.5 eq.), NH ₄ I (1.5 eq.)	59
4	Et ₄ NI (2.0 eq.), NH ₄ I (0.5 eq.)	47
5	Et ₄ NI (2.0 eq.), NH ₄ I (1.5 eq.)	71
6	Et ₄ NI (2.0 eq.), -	62
5	-, NH ₄ I (2.0 eq.)	trace
7	Et ₄ NI (2.0 eq.), LiI (2.0 eq.)	75

Question 4. How to react under argon gas conditions?

Response: Thanks for the comments. We have conducted this transformation under argon atmosphere and gave the desired product **2a** in 78% yield (**Fig. R5**).

Fig. R5

Reviewer 3

Question 1: For products originating from internal alkenes, kindly specify the stereochemistry of the predominant product. Additionally, clarify if cis- and trans-alkenes yield the same or different products.

Response: Thanks for the comments. We have confirmed the configuration of product **2y** via single-crystal X-ray diffraction analysis (**Fig. R6**). The analysis of the resulting single crystal elucidated that the main stereoselectivity of internal alkenes was a trans configuration of the compound **2y**. The above results confirmed the procedure of epoxy ring-opening in the mechanism: the nucleophilic reagent (H₂O) attacked protonated epoxy from the opposite direction of the ring to obtain the corresponding trans ring-opening product.

Compound **2y** (25 mg) was dissolved in 6 mL of ether/n-hexane ($v_1/v_2 = 1:2$), and it was crystallized to give crystal as yellow prisms after the solvent was slowly volatilized in 7 days at room temperature (~ 28 °C).

All diffraction data were obtained on a Bruker Smart Apex CCD diffractometer equipped with graphite-monochromated Mo K α radiation. CCDC-2286663 (**2y**), contain the supplementary crystallographic data. These data can be obtained free of charge from the Cambridge Crystallographic Data Centre (<http://www.ccdc.cam.ac.uk/>). X-ray crystallographic data for **2y** is available as **Table R2**.

Fig. R6. The molecular structure of **2y**

Table R2. Crystal data and structure refinement for 2y

Empirical formula	C ₁₉ H ₂₁ BrO ₃	
Formula weight	377.27	
Temperature/K	293	
Crystal system, Space group	monoclinic, P21/c	
Unit cell dimensions	a/Å 24.4551(9)	α/° 90
	b/Å 8.2654(4)	β/° 90.774(3)
	c/Å 8.4350(3)	γ/° 90
Volume/Å ³	1704.82(12)	
Z	4	
ρ _{calc} /cm ³	1.470	
μ/mm ⁻¹	3.387	
F(000)	776.0	
Crystal size/mm ³	0.07 × 0.06 × 0.05	
Radiation	CuKα (λ = 1.54184)	
2θ range for data collection/°	3.614 to 136.82	
Index ranges	-28 ≤ h ≤ 20, -9 ≤ k ≤ 8, -10 ≤ l ≤ 8	
Reflections collected	8664	
Independent reflections	3055 [R _{int} = 0.0245, R _{sigma} = 0.0254]	
Data/restraints/parameters	3055/0/211	
Goodness-of-fit on F ²	1.068	
Final R indexes [I ≥ 2σ (I)]	R ₁ = 0.0321, wR ₂ = 0.0873	
Final R indexes [all data]	R ₁ = 0.0372, wR ₂ = 0.0909	
Largest diff. peak/hole / e Å ⁻³	0.24/-0.39	

In addition, we conducted the reaction using cis and trans dienes as the starting material (**1w** and **1w'**) for the synthesis of compound **2w** and **2w'** under standard conditions (**Fig.R7**). The above results showed that the yield of the corresponding diol **2w** from trans alkene (**1w**, 64%) is slightly higher than that from cis alkene (**2w'**, 58%). Furthermore, the structures of products **2w** (*d.r.* > 20:1) and **2w'** (*d.r.* > 20:1) were confirmed by nuclear magnetic resonance (NMR) spectroscopy, which indicating that **2w** and **2w'** were different in structure. By the way, we have revised the structure of compound **2w** in our manuscript. To demonstrate the good generality of the reaction, we added several substrates, so we changed the substrate number and highlighted it in yellow in our manuscript.

Fig.R7

¹H NMR of Compound 2w (400 MHz, CDCl₃):

¹³C NMR of Compound 2w (100 MHz, CDCl₃):

¹H NMR of Compound 2w' (400 MHz, 400 MHz, CDCl₃):

¹³C NMR of Compound 2w' (100 MHz, CDCl₃):

Question 2: The experiments demonstrate that electricity is required to convert intermediate **4** into the desired products. However, the proposed mechanism does not account for this observation. Please revise the mechanism accordingly to incorporate this crucial aspect.

According to the control experiments and literatures made us to refine the reaction mechanism (**Fig. R4**). Firstly, electrochemical oxidation of I⁻ generated iodine radical, which then dimerized to generate iodine. Then, iodine reacted with alkene to generate iodonium intermediate **Int-I**. **Int-I** tended to engage in the nucleophilic addition with water to generate intermediate **Int-II**. Meanwhile, Et₄NI was decomposed into Et₃N on the anode, which was then electrooxidized to obtain the α-amino radical. Then, the α-amino radical was capable of abstracting iodine atom from alkyl iodides (intermediate **Int-II**) to give **Int-III** via intramolecular cyclization. Subsequently, a ring-opening reaction occurred of **Int-III** by the attack of H₂O furnishing the products. Simultaneously, an equilibrium reaction between **Int-III** and free epoxy intermediate (**Int-IV**) proceeded in this transformation.

Fig. R4 Proposed Mechanism

By the way, to demonstrate the good generality of the reaction, we added several substrates (**Fig. R8**), so we changed the substrate number and highlighted it in yellow in our manuscript. And the corresponding data have been added to the manuscript and supporting information.

Fig. R8

REVIEWERS' COMMENTS

Reviewer #1 (Remarks to the Author):

The authors have effectively addressed the issues raised by the reviewers. The quality of the manuscript has significantly improved, and I believe it is now suitable for acceptance and publication.

Reviewer #2 (Remarks to the Author):

The authors have addressed my confusion and this paper can now be accepted.

Reviewer #3 (Remarks to the Author):

The authors have successfully addressed the previous comments. However, there is an additional point to consider. In the newly proposed mechanism, the transition from int-II to int-III necessitates the loss of an electron. The process by which this occurs remains unclear.

Comments from the reviewers:

To reviewer 1:

Reviewer letter: The authors have effectively addressed the issues raised by the reviewers. The quality of the manuscript has significantly improved, and I believe it is now suitable for acceptance and publication.

Answer: We highly appreciate the reviewer's positive comments on our manuscript.

To reviewer 2:

Reviewer letter: The authors have addressed my confusion and this paper can now be accepted.

Answer: We highly appreciate the reviewer's positive comments on our manuscript. We are sure that the quality of this work has been greatly improved according to this nice comments and wise suggestions. Thanks very much.

To reviewer 3:

Reviewers letter: The authors have successfully addressed the previous comments. However, there is an additional point to consider. In the newly proposed mechanism, the transition from Int-II to Int-III necessitates the loss of an electron. The process by which this occurs remains unclear.

Answer: Thanks for the comments. We are so sorry for the mistake, we have modified the reaction mechanism: Firstly, electrochemical oxidation of I⁻ generated iodine radical, which then dimerized to generate iodine. Then, iodine reacted with alkene to generate iodonium intermediate **Int-I**. **Int-I** tended to engage in the nucleophilic addition with water to generate intermediate **Int-II**. Meanwhile, Et₄NI was decomposed into Et₃N on the anode, which was then electrooxidized to obtain the α-amino radical. Then, the α-amino radical was capable of abstracting iodine atom from alkyl iodides (intermediate **Int-II**) to give **Int-III** via intramolecular cyclization. Simultaneously losing an electron during this process. Subsequently, a ring-opening reaction occurred of **Int-III** by the attack of H₂O furnishing the products. Simultaneously, an equilibrium reaction between **Int-III** and free epoxy intermediate

(Int-IV) proceeded in this transformation.

Figure R1 Possible mechanisms